# Depression and Medicine Use among Older Adults during the COVID-19 Pandemic: The Role of Psychosocial Resources and COVID-19 Perceived Susceptibility

**DOI:** 10.3390/ijerph20043398

**Published:** 2023-02-15

**Authors:** Lee Greenblatt-Kimron, Shiri Shinan-Altman, Mordechai Alperin, Inbar Levkovich

**Affiliations:** 1School of Social Work, Ariel University, Ariel 4070000, Israel; 2Louis and Gabi Weisfeld School of Social Work, Bar Ilan University, Ramat Gan 5290002, Israel; 3Department of Family Medicine, The Ruth & Bruce Rappaport Faculty of Medicine, Technion-Israel Institute of Technology, Haifa 3200003, Israel; 4Clalit Health Services, Haifa and Western Galilee District, Tel Aviv 6209804, Israel; 5Faculty of Graduate Studies, Oranim Academic College of Education, Kiryat Tivon 3600600, Israel

**Keywords:** depression, medication use, optimism, social support, COVID-19 perceived susceptibility

## Abstract

A relationship was found between the COVID-19 pandemic and depression among older adults and between depressed mood and increased use of antidepressant medication among older adults during the pandemic. With the aim of broadening the understanding of these relationships, the study examined whether COVID-19 perceived susceptibility mediates the relationship between psychosocial resources (optimism and perceived social support) and depressive symptoms and medication use. Participants included 383 older adults (*M* = 71.75, *SD* = 6.77) reporting on socio-demographics, health characteristics, depression, optimism, social support, and COVID-19 perceived susceptibility. Medication use was retrieved from participants medical files. Lower optimism, lower social support, and higher COVID-19 perceived susceptibility were associated with greater depression, related with higher medication use. The findings emphasize the buffering effect of psychosocial resources on the adverse effects of depression affecting older adults during the COVID-19 pandemic, and consequently, the increased use of medication in this population. Practitioners should focus interventions on enhancing optimism and expanding social support among older adults. Moreover, interventions focused on alleviating depression among older adults should aim at improving perceptions of perceived susceptibility in the older population.

## 1. Introduction

Severe acute respiratory syndrome coronavirus-2 (SARS-CoV-2), the virus accountable for the COVID-19 pandemic, has extensively impacted the worldwide population [1]. Old age was affirmed as a risk for COVID-19 complications [2]; consequently, policymakers and media instructed older adults to confine social interactions [3]. Depression is a prevalent mental disorder among older adults generating distress and reduced quality of life [4]. Links were reported between the COVID-19 pandemic and depression among older adults [2]. Older age was established as a threat for COVID-19 complications [2], with studies during the pandemic showing that older adults displayed depressive symptoms, anxiety [5], peritraumatic distress [6], post-traumatic stress disorder [7], and depression even after receiving the vaccination [8]. Moreover, a link was found between loneliness with depressed mood and increased use of antidepressant medication among older adults during the first COVID-19 lockdown in the UK [9]. Nevertheless, factors exploring adverse mental reactions together with an increase in medicine use among the older population during pandemics is limited. The American Geriatrics Society recently stated that knowledge on the benefits and harmful outcomes of medication use in older adults is frequently limited [10]; therefore, factors exploring increased medication use in this population is vital, particularly during stressful life events such as the COVID-19 pandemic. On this basis, the present study aimed to shed light on variables predicting depression and medicine use among older adults during the pandemic. Specifically, the study aimed to examine the contribution of psychosocial resources (optimism and social support, respectively) and COVID-19 perceived susceptibility to predict depression and medication use among older adults.

Depressive symptoms are perilous to older adults’ overall health [4], with studies reporting links between the COVID-19 pandemic and depression in older adults [2]. Beck’s [11] cognitive developmental model of depression proposes that depressive symptoms embody negative interpretations of life events as demonstrated during the pandemic, with a relationship found between older adults holding negative world assumptions and a 4.4 times higher probability for clinical depressive symptoms than those with positive world assumptions [8]. Of particular significance to the current study, psychosocial resources, namely optimism and social support, were found to buffer depression among older adults during the pandemic [12]. Likewise, lower COVID-19 perceived susceptibility was linked with lower depression among older adults [12]. Nevertheless, these effects have been less explored in terms of medication use among older adults.

Optimism and pessimism play a vital role when coping with negative experiences [13], such as the COVID-19 pandemic as optimistic people strive when confronted with life’s challenges, while those with a more pessimistic outlook tend to withdraw [14]. Optimism, known as mechanism for appraising one’s life [15], may be essential in older age. Studies show relationships between optimism in older adults and improved self-reported health [16], positive emotions [17], and life satisfaction [15]. During the pandemic, optimism and social support were found to buffer depression among older adults [12]. Nevertheless, there is need for further study of this association among older adults on physical and mental health [12], specifically, medication use during stressful life events such as the COVID-19 pandemic. Social support is a buffer from the adverse outcomes of distress on mental and physical health [18]. Indeed, a systematic review of 24 studies among community-dwelling older adults found a relationship between adequate social support and more minor depressive symptoms [19]. On this basis, the social distancing and self-isolation policies of the COVID-19 pandemic may have adverse mental consequences due to the absence of social support [20], as demonstrated by studies around the globe during the COVID-19 pandemic. For example, a German study showed a link between perceived social support and lower levels of anxiety, depression, and sleeping disorders [21,22,23]. This relationship will be explored in the current study in the context of depression and medication use during the COVID-19 pandemic among older adults.

Perceived susceptibility describes the subjective threat of contracting a disease [24], which is known to be associated with increased depression in later life, especially during uncertain situations [5]. Perceived susceptibility also plays a part in health protection behaviors such as wearing face masks [25] and is, therefore, significant during epidemics [26]. Although health awareness increased during the COVID-19 pandemic [27], a lower perceived susceptibility of contracting COVID-19 was linked with fewer protective behaviors [28]. Studies during the pandemic linked perceived susceptibility to COVID-19 and mental well-being, demonstrated by an association between social closeness to people infected by COVID-19 with higher perceived susceptibility and higher anxiety levels [29], such as depression among older adults [12]. Nevertheless, this association needs further exploration among older adults, while the relationship between perceived susceptibility and medicine use among older adults remains unexplored.

Based on the above, the current study aimed to broaden the knowledge on the mental outcomes of stressful life events in old age, specifically factors associated with depression and the use of medication among older adults during the COVID-19 pandemic. The first study hypothesis maintained that higher optimism and higher perceived social support will be negatively associated with depression, while higher COVID-19 perceived susceptibility will be positively associated with depression, which in turn will be positively associated with more use of medication among older adults. The second study hypothesis maintained that perceived susceptibility would mediate the relationship between psychosocial resources (optimism and perceived social support) and depressive symptoms and medication use.

## 2. Materials and Method

### 2.1. Research Population and Sample

A total of 383 older adults participated in the study. See Table 1 for a detailed description of socio-demographic and health characteristics as well as use of medication. The Institutional Review Board of Clalit Health Services approved the study (approval number 0060-20-COM2). Data were collected in June and July 2020, during which the first wave of COVID-19 subsided in Israel. Over this period, Israel was under strict social distancing regulations. The third researcher (M-A) recruited nine family medicine interns willing to distribute the questionnaires and interview the participants by telephone. The interns received training on conducting an interview from the researchers (S.S. and A.L.) who supervised the conduct of the interviews. Participation was on a voluntary basis. The response rate was high, with only approximately 5% of potential participants choosing not to participate in the study. The findings support the cognitive model of coping with stressors. Participants signed an informed consent form before commencing participation. The list of sleep medications and antidepressants used by the participants was retrieved from their files, including Benzodiazepine drugs, Antidepressant SNRIs, and other antidepressants (Miro 45 TAB 45 mg 30, Bonserin (Bolvidon) TAB 30 mg 20, Trazodil TAB 100 mg 30, Remotiv TAB 500 mg 30).

### 2.2. Measures

Socio-demographic and health characteristics included gender, age, years of education, marital status, number of children with whom participants live, and sources of help during COVID-19. Participants reported health characteristics including perceived health status (1 = excellent, 5 = very bad) and other types of chronic diseases (such as hypertension, diabetes, coronary heart disease, cancer, lung disease, diabetes, hypertension, other).

Depression was measured using the Symptoms of Depression Questionnaire (Center for Epidemiological Studies Depression, CESD–10) [30]. Participants were asked to rate the intensity of their experiences during the last week on a 4-point Likert-type scale, ranging from 0 (never) to 3 (to a great extent). After reversing the negative statements, the mean score was calculated, with higher scores reflecting a higher level of depression (Cronbach’s α = 0.85). The cutoff score for clinical classification is defined as 10 (in a range of 0 to 30) [30].

Optimism was measured using the Life Orientation Test (LOT–R) [31]. This instrument is a six-item scale, with three items worded as positive statements (e.g., “In uncertain times, I usually expect the best”) and the other three as negative statements (e.g., “If something can go wrong for me, it will”), which reflect a patient’s expectations regarding the future. Participants were asked to indicate the extent to which they agreed or disagreed with each item on a 5-point Likert-type scale ranging from 1 (completely disagree) to 5 (completely agree). After reversing the negative statements, a mean score was calculated, with higher scores indicating a higher level of optimism (Cronbach’s α = 0.64).

Social support was measured using the Multidimensional Scale of Perceived Social Support [32]. Using this 12-item scale, participants were asked to indicate the extent to which they agreed or disagreed with each item on a 5-point Likert scale ranging from 1 (strongly disagree) to 5 (strongly agree) (e.g., “My family really tries to help me”). The mean score was calculated; a high score indicated greater levels of perceived social support (Cronbach’s α = 0.90).

COVID-19 perceived susceptibility was assessed based on previous studies conducted among the lay public [33] using a single item: “What do you think is the likelihood that you will contract COVID-19?” Answers were rated on a 5-point Likert-type scale, ranging from 1 (negligible) to 5 (very high). Variable distribution did not deviate from a normal distribution (skewness = 0.21, *SE* = 0.12) and it was used as normally distributed.

### 2.3. Data Analysis

Data were analyzed using SPSS version 27. Descriptive statistics provided information about participants’ demographic characteristics and the research variables. Pearson correlations were calculated to assess the associations between the research variables. *t*-tests and Pearson correlations were calculated for the relationships between depression and the participants’ demographic characteristics. *t*-tests, a simple logistic regression, and the z-ratio for the significance of the difference between two independent proportions, were calculated for the relationships between medication use and the participants’ demographic characteristics. A multiple hierarchical regression was calculated for depression, and a logistic regression for medication use. Gender, age, and years of education were entered as control variables in the first step, optimism and social support in the second step, and perceived susceptibility in the third step.

## 3. Results

As shown in Table 2, mean depression was moderate-low, and 38.9% (*n* = 149) of the participants were classified as suffering from depression. About 31% of the participants (*n* = 119) were using sleep medications and/or antidepressants. The joint occurrence of depression and the use of sleep medications and/or antidepressants was significant (*Z* = 2.88, *p* = 0.004). In addition, the total score for depression was significantly higher among those using sleep medications and/or antidepressants (*M* = 1.14, *SD* = 0.66) than among those not using them (*M* = 0.83, *SD* = 0.60) (*t* (381) = 4.47, *p* < 0.001). In general, 80 participants (20.9%) were using Benzodiazepine medications, 25 participants (6.5%) were using SNRIs or other antidepressants, and 14 participants (3.7%) were using both Benzodiazepine medications and SNRIs or other antidepressants.

Depression, optimism, and COVID-19 perceived risk were not different by the type of medication used. However, social support was higher among participants using Benzodiazepine medications (*M* = 4.02, *SD* = 0.72) than among others (SNRIs: *M* = 3.84, *SD* = 0.80, SNRIs and Benzodiazepine medications: *M* = 3.61, *SD* = 0.92) (*F*(2, 114) = 5.81, *p* = 0.004, η^2^ = 0.096). In addition, as mentioned above, depression was higher among those using any medication than not using, yet no difference was found for the other study variables by medication use.

As further shown in Table 2, depression was higher among women (*M* = 1.04, *SD* = 0.66) than men (*M* = 0.80, *SD* = 0.59) (*t* (381) = 3.71, *p <* 0.001), and was negatively related with years of education (*r* = −0.25, *p* < 0.001). Participants using medication were older (*M* = 73.15, *SD* = 7.26) than others (*M* = 71.13, *SD* = 6.46) (*t* (381) = 2.70, *p* = 0.007). These demographic variables were controlled for in further analyses. In addition, depression was negatively related with perceived health status (*r* = −0.48, *p* < 0.001, *n* = 311), but was unrelated with the existence of a chronic disease (*p* = 0.070, *n* = 339). The use of medication was negatively related with perceived health status (*OR* (95% *CI)* = 0.75 (0.57, 0.98), *p* = 0.038, non-use of medication: *M* = 2.73 *SD* = 0.94, medication use: *M* = 2.50, *SD* = 0.85), and was positively related with the existence of a chronic disease (*Z* = 2.83, *p* = 0.005, *n* = 97 33.2% of those with a chronic disease, *n* = 6 12.8% of those without a chronic disease).

A multiple regression analysis was calculated for depression, and a logistic regression for medication use. Gender (1—male, 0—female), age, and years of education served as control variables, optimism, social support, and perceived susceptibility were the independent variables. An attempt was made to include perceived health status and the existence of a chronic disease in the regression models, due to their relationships with the dependent variables. Their contribution was found to be non-significant, and sample size was reduced due to their missing data. Thus, they were excluded from the following regression models, presented in Table 3.

Both regression models were found to be significant, with 19% of the explained variance in depression and about 5% in medication use. Depression, as noted earlier, was higher for women, and for participants with lower levels of education. Optimism and social support added 7% to the explained variance in depression, beyond the demographic variables, and perceived susceptibility added another 2% to the total explained variance in depression. Lower optimism, lower social support, and higher perceived susceptibility were associated with greater depression. Medication use was found only to be related with age, such that older participants were at a higher risk for using Benzodiazepine medications, SNRIs, or other antidepressants. Finally, perceived susceptibility was not found to mediate the relationship between optimism and social support and depression or mediation use. Both optimism and social support were unrelated with perceived susceptibility (*r* = −0.04, *p* = 0.389, and *r* = −0.09 *p* = 0.079, respectively), and perceived susceptibility was unrelated with medication use (*r* = 0.01, *p* = 0.939), even though it was related with depression (*r* = 0.17, *p* < 0.001). Thus, depression was directly related with gender, years of education, optimism, social support, and perceived susceptibility. Medication use was related with the participants’ age, as well as with depression.

## 4. Discussion

The present study is the first to examine the relationship between the personal resources of optimism and social support with depression and medicine use among older adults during the COVID-19 pandemic, and in particular, the mediating role of COVID-19 perceived susceptibility in this relationship. About 39% of the participants reached a clinical level for depression, with around 31% using medications for anxiety, depression, or sleep. The high depression rate was similar to a previous study of Israeli older adults during the COVID-19 pandemic, reporting 37.5% of depression in the study sample. Nevertheless, these figures were higher than in study that examined late-life depression across Europe using data from the Survey on Health, Aging and Retirement in Europe (SHARE) between the years 2004–2015, i.e., before the start of the pandemic. The researchers found depression in 35% of older adults in Southern Europe, 32% in Central and Eastern Europe, 26% Western Europe, and 17% in Scandinavia [34], which may indicate increased depression levels during the pandemic among older adults. In terms of demographic characteristics, depression was higher for older adult women than man, supporting a previous study on depression among older adults during the COVID-19 pandemic [12]. This finding also supports previous studies before the COVID-19 pandemic, showing a higher prevalence of depressive symptoms among older adult women compared to older adult men [35]. The link between depression with lower education in the current study is also in line with previous studies, e.g., [36]. The use of medication in the current study use was related to age, with older participants reporting higher use of Benzodiazepine medications, SNRIs, and other antidepressants. The results in the current study may be explained by findings of previous studies, showing a relationship between mental health problems at times of crises and traumatic events with pharmacologic treatment such as benzodiazepines, e.g., [37].

In line with the first hypothesis, higher optimism was found to be negatively related to depression. This association highlights the buffering role of psychological resources during stressful life events. The findings of the current study are in line with the postulation that optimism is essential for coping with negative experiences [13] (i.e., the COVID-19 pandemic) and support the suggestion that optimistic people endeavor when confronted with adverse events, while those who are more pessimistic tend to recoil [14]. Moreover, the link between higher optimism and lower levels of depression is especially pertinent for older adults as optimism is a known mechanism for appraising one’s life [15]. The current results also coincide with previous studies during the COVID-19 pandemic, reporting links between higher levels of optimism and less depression among older adults [12].

Confirming the first hypothesis, social support was negatively associated with depression. This finding underscores the importance of social support for older adults during the COVID-19 pandemic, as the adverse mental outcomes associated with the absence of social support have been highlighted as a characteristic of the COVID-19 pandemic, especially due to the social distancing and self-isolation policies [20]. Moreover, older adults may find isolation more troublesome than their younger counterparts, therefore, requiring more support [12]. This is essential, as studies from the COVID-19 pandemic have shown links between perceived social support and fewer adverse outcomes including depression [12,21] and sleeping disorders [21].

Additionally, consistent with the first hypothesis, higher COVID-19 perceived susceptibility was found to be positively related with depression. This finding is in line with previous studies showing that perceived susceptibility increases depression among older adults, particularly in uncertainty situations [5,12], especially for those in quarantine, in high-risk locations, and social closeness to others infected by the COVID-19 virus [29]. A previous study reported a link between perceived susceptibility of contracting COVID-19 with fewer protective behaviors [28], which may be crucial for older adults as older age is known as a risk for COVID-19 complications [2].

Congruous to the first hypothesis a link was shown between depression and use of medication among older adults. Specifically, older adults who reported higher levels of depression also reported more use of Benzodiazepine medications, SNRIs or other antidepressants. This finding supports a previous study reporting a link between loneliness with depressed mood and increased use of antidepressant medication among older adults during the first COVID-19 lockdown in the UK [9].

In contrast to the second hypothesis, perceived susceptibility was not found to mediate the relationship between optimism and social support and depression or mediation use. Moreover, contrasting the results of a previous study among older adults [12], optimism and social support were unrelated with perceived susceptibility. Although these findings were unexpected, they may be explained by the Health Belief Model (HBM) by which preventive health behavior transformations are initially based on six facets, namely, susceptibility, seriousness, benefits, barriers, health motivation, and confidence that one can successfully execute behavior with a positive outcome, which appears to be particularly relevant among older adults during the COVID-19 pandemic [38].

Nevertheless, the current study has several limitations. First, it used a cross-sectional design; therefore, causality cannot be concluded from the current findings. Second, the study is based on self-report questionnaires, which may bias the participants’ responses. Third, although the response rate was high, there may be sampling bias due to physical difficulties such as answering the phone or hearing deficits. Fourth, the study had a low sample size, and the results may be influenced by the regional and cultural characteristics of the sample (i.e., Israel). Fifth, more than one medical intern interviewed the participants; therefore, it can be determined if the manner in which they interviewed the participants affected their answers. Finally, due to the different waves and the vaccination programs, longitudinal research is needed to determine the long-term effects of the pandemic among older adults, particularly on adverse mental and physical outcomes such as depression and medication use.

## 5. Conclusions

Despite these limitations, the study emphasizes that psychosocial resources and COVID-19 perceived susceptibility impact the adverse effects of depression and medicine use in the older population. This suggests that when psychosocial resources are both internal (optimism) and external (social support), it may be easier to buffer these effects, thus allowing older adults to feel calmer and cope adequately with the stress of COVID-19. On a practical level, the rise in depression among older adults since the COVID-19 outbreak cannot be underscored [12]. In this regard, the findings of the current study provide evidence for the need of suitable interventions that enhance psychosocial resources and direct COVID-19 perceived susceptibility to appropriate health behaviors with the aim of reducing the long-lasting effects of depression and medication use in the older population, with a special attention to those with a low education level. Mental health practitioners should promote psychosocial resources in this population through both individual and community interventions. On the community level, mental health practitioners should implement both frontal and online community-based projects to enhance older adults’ sense of perceived social support. 

## Figures and Tables

**Table 1 ijerph-20-03398-t001:** Participants’ demographic characteristics (*N* = 383).

Health Characteristics	*N* (%)
Gender (%)	
Male	180 (47.0)
Female	203 (53.0)
Mean age (*SD*), range	71.75 (6.77), 60–95
Marital status (%)	
Married	284 (74.2)
Not married	99 (25.8)
Mean number of children (*SD*), range	3.03 (1.25), 0–10
Mean number of years of education (*SD*), range	12.98 (3.04), 6–24
Living with (%)	
Alone	64 (17.3)
Intimate partner	262 (70.6)
Family member/s	42 (11.3)
Formal caregiver	3 (0.8)
Perceived health status (%)	
Very bad	42 (11.0)
Bad	140 (36.6)
Moderate	116 (30.3)
Good	55 (14.4)
Excellent	30 (7.8)
Chronic disease (%)	
Yes	292 (86.1)
No	47 (13.9)
Types of other chronic diseases (%)	
Hypertension	97 (33.2)
Diabetes	27 (9.2)
Coronary heart disease	33 (11.3)
Cancer	36 (12.3)
Lung disease	17 (5.8)
Diabetes and hypertension	63 (21.6)
Other	19 (6.5)
Mean Depression (*SD*), range	9.28 (6.36), 0–30
Depression cutoff–positive (%)	149 (38.9)
Medication for anxiety, depression, sleep (%)	119 (31.1)

**Table 2 ijerph-20-03398-t002:** Means, Standard deviations, Ranges, and Pearson Correlations for the Study Variables (*n* = 383).

Variables	Gender	Age	Education Years	Optimism	Social Support	Perceived Susceptibility	Depression	Medication (Yes)
1. Gender	-							
2. Age	0.06	-						
3. Education years	−0.04	−0.04	-					
4. Optimism	0.11 *	0.02	−0.05	-				
5. Social support	−0.07	−0.13 *	0.21 ***	0.14 **	-			
6. Perceived susceptibility	−0.11 *	0.02	−0.01	−0.04	−0.09	-		
7. Depression	−0.19 ***	0.08	−0.25 ***	−0.20 ***	−0.29 ***	0.17 ***	-	
8. Medication (yes)	−0.06	0.14 **	−0.08	0.02	−0.05	0.01	0.22 ***	-
Mean	0.47	71.75	12.98	3.57	4.11	2.52	0.93	0.31
SD	0.50	6.77	3.04	0.72	0.73	0.96	0.64	0.46
Possible range				1–5	1–5	1–5	0–3	0–1
Actual range		60–95	6–24	1.3–5	1–5	1–5	0–2.75	0–1

* *p* < 0.05, ** *p* < 0.01, *** *p* < 0.001.

**Table 3 ijerph-20-03398-t003:** Multiple hierarchical regressions for depression and medication use (*n* = 383).

	Depression				Medication Use			
	*B*	*SE*	β	Adj. *R*^2^	*B*	*SE*	OR (95% CI)	Nagelkerke’s *R*^2^
Step 1				0.10 ***				0.05 **
Gender	−0.25	0.06	−0.20 **		−0.32	0.23	0.72 (0.46, 1.14)	
Age	0.01	0.01	0.08		0.05	0.02	1.05 ** (1.01, 1.08)	
Education years	−0.05	0.01	−0.25 ***		−0.06	0.04	0.94 (0.87, 1.01)	
Step 2				0.17 ***				0.05 *
Gender	−0.24	0.06	−0.19 ***		−0.35	0.23	0.71 (0.45, 1.12)	
Age	0.01	0.01	0.06		0.04	0.02	1.05 ** (1.01, 1.08)	
Education years	−0.05	0.01	−0.22 ***		−0.06	0.04	0.95 (0.88, 1.02)	
Optimism	−0.14	0.04	−0.16 ***		0.11	0.16	1.11 (0.81, 1.53)	
Social support	−0.18	0.04	−0.21 ***		−0.11	0.16	0.90 (0.65, 1.24)	
Step 3				0.19 ***				0.05 *
Gender	−0.22	0.06	−0.17 ***		−0.36	0.24	0.70 (0.44, 1.11)	
Age	0.01	0.01	0.06		0.04	0.02	1.05 ** (1.01, 1.08)	
Education years	−0.05	0.01	−0.22 ***		−0.06	0.04	0.95 (0.88, 1.03)	
Optimism	−0.14	0.04	−0.16 ***		0.10	0.16	1.11 (0.81, 1.52)	
Social support	−0.18	0.04	−0.20 ***		−0.11	0.16	0.89 (0.65, 1.23)	
Perceived susceptibility	0.08	0.03	0.12 **		−0.06	0.12	0.94 (0.74, 1.20)	

* *p* < 0.05, ** *p* < 0.01, *** *p* < 0.001. Depression: *F* (6, 376) = 14.82, *p* < 0.001; Medication use: χ^2^(6) = 12.96, *p =* 0.044.

## Data Availability

None.

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
