# Peer review of "Depression and Medicine Use among Older Adults during the COVID-19 Pandemic: The Role of Psychosocial Resources and COVID-19 Perceived Susceptibility"

_ijerph, 2023, doi:10.3390/ijerph20043398_

Round 1

Reviewer 1 Report

 The study needs a wide revision, especially in scientific writing and fluency between paragraphs. Redundancy, excessive use of prepositions, and long sentences are some of the points that require improvement. In terms of innovation, the introduction does not make it clear which scientific gaps on the subject still exist and their importance in the scientific scenario.

Introduction

The introduction is unnecessarily long, separating readers from the study’s main purpose and its innovation and scientific contribution. I recommend removing all subtopics from this section and limiting it to 4-5 paragraphs. Gather the information generated so far on the subject and list the innovation of the study in the last paragraph, highlighting the objective and hypothesis of the study.

Methods

Line 133 - I do not agree with decimal places for age, but if the authors want to keep it, I would like an explanation of its meaning.

Lines 132 – 139 – Do not repeat the information presented in Table 1.

Lines 141 – 144 – These sentences are redundant.

Lines 155 – 156 – Present some data validating this instrument.

Lines 179 – 183 – Same thing here.

Line 185 – ver?

Line – 185 – “Descriptive” and “Describe”. This is one example of the improvement that the manuscript requires.

Results

This entire section needs to be reformulated. It is extremely polluted and avoids readers to reach the main results. Limit the text to the most important outcomes.

The layout of Table 2 is confusing. I recommend inserting the variable’s names also in the first line.

Discussion

Do not repeat the results in this section.

Lines 272 – 275 – Avoid the term “To the best of our knowledge”. Also, without presenting the data that validates “perceived susceptibility”, this statement becomes nonscientific.  Furthermore, the authors must be aware of their low sample size for this type of study. I recommend limiting this statement (along with the introduction and conclusion) to the regional characteristics of the sample (i.e Israel) since the observed results are likely to be influenced by geographic and cultural perspectives.

Lines 350 – 351 – Provide the reference for this statement. This point must be carefully considered by the authors. Some volunteers may provide non-accurate answers while interviewed “face to face” because of the private information required in some depression questionnaires.

Conclusion

Limit the conclusion to one short paragraph.

Adjust reference 45.

Author Response

Reviewer 1: The study needs a wide revision, especially in scientific writing and fluency between paragraphs. Redundancy, excessive use of prepositions, and long sentences are some of the points that require improvement. In terms of innovation, the introduction does not make it clear which scientific gaps on the subject still exist and their importance in the scientific scenario.

Answer: We thank the Reviewer for the helpful suggestions for revising and improving our manuscript. We revised the paper with the aim of writing it in a more concise manner and made corrections as suggested by the Reviewer as follows:

Introduction

Reviewer 1: The introduction is unnecessarily long, separating readers from the study’s main purpose and its innovation and scientific contribution. I recommend removing all subtopics from this section and limiting it to 4-5 paragraphs. Gather the information generated so far on the subject and list the innovation of the study in the last paragraph, highlighting the objective and hypothesis of the study.

Answer: We thank the Reviewer for this significant suggestion. We have deleted all subtopics and shortened all the paragraphs as suggested by the Reviewer, In addition, we highlighted the objective in the last paragraph. We have also rearranged the references in accordance.

Methods

Reviewer 1: Line 133 - I do not agree with decimal places for age, but if the authors want to keep it, I would like an explanation of its meaning.

Answer: This was deleted.

Reviewer 1: Lines 132 – 139 – Do not repeat the information presented in Table 1.

Answer: The information was removed and a reference was made to the Table.

Reviewer 1: Lines 141 – 144 – These sentences are redundant.

Answer: This was corrected to avoid redundancy.

Reviewer 1: Lines 155 – 156 – Present some data validating this instrument.

Answer: This information was added.

Reviewer 1: Lines 179 – 183 – Same thing here.

Answer: This information was added.

Reviewer 1: Line 185 – ver?

Answer: This was corrected.

Reviewer 1: Line – 185 – “Descriptive” and “Describe”. This is one example of the improvement that the manuscript requires.

Answer: This was corrected.  

Results

Reviewer 1: This entire section needs to be reformulated. It is extremely polluted and avoids readers to reach the main results. Limit the text to the most important outcomes.

Answer: Thank you for this important suggestion. We have reformatted the results highlighting only the most important outcomes.

Reviewer 1: The layout of Table 2 is confusing. I recommend inserting the variable’s names also in the first line.

Answer: We have added the variable’s names also in the first line. 

Discussion

Reviewer 1: Do not repeat the results in this section.

Answer: These were deleted.

Reviewer 1: Lines 272 – 275 – Avoid the term “To the best of our knowledge”. Also, without presenting the data that validates “perceived susceptibility”, this statement becomes nonscientific.  Furthermore, the authors must be aware of their low sample size for this type of study. I recommend limiting this statement (along with the introduction and conclusion) to the regional characteristics of the sample (i.e Israel) since the observed results are likely to be influenced by geographic and cultural perspectives.

Answer: The term “To the best of our knowledge” was deleted. The low sample size and regional and cultural characteristics of the sample were added to the limitations of the study.

Reviewer 1: Lines 350 – 351 – Provide the reference for this statement. This point must be carefully considered by the authors. Some volunteers may provide non-accurate answers while interviewed “face to face” because of the private information required in some depression questionnaires.

Answer: In accordance with this comment, we deleted this statement.

Conclusion

Reviewer 1: Limit the conclusion to one short paragraph.

Answer: We have adjusted this is accordance.

Reviewer 1: Adjust reference 45.

Answer: This was adjusted.

Finally, numbers of all references cited in the text and the reference list were corrected in accordance with the changes suggested by both Reviewers.

We once again thank the Reviewer for the valued comments that helped improve our paper. We hope that it is now ready for acceptance.

*******************************************************

Reviewer 2 Report

This paper proposes two hypotheses related to older adults, COVID-19, and depression, (1) “higher optimism and higher perceived social support will be negatively associated with depression, while higher COVID-19 perceived susceptibility will be positively associated with depression, which in turn will be positively associated with more use of medication among older adults” and (2) “perceived susceptibility would mediate the relationship between psychosocial resources (optimism and perceived social support) and depressive symptoms and medication use.” To test these hypotheses, 383 older Israeli adults were interviewed by phone by medical interns to answer a series of questions constructed by the authors based on previous research in this regard. The results were that the first hypothesis was confirmed, but the second was not. The authors speculate that the reason for the unexpected result regarding the second hypothesis may be explained by the Health Belief Model. The study is seen to have both theoretical and practical implications related to psychosocial resources affecting older adults. It is considered the first study of its kind with respect to older adults.

The strengths of this manuscript are that it is clearly written, each of the measures used is properly referenced, the measures chosen were reasonable and directly relevant to the hypotheses to be investigated and the work is in an under studied area in need of research. The weaknesses are that the authors base important aspects of their research on old references without an investigation into whether the research conducted at that time still holds true today. This work will have to be done to improve the paper. Furthermore, although this is a paper directly related to COVID-19 and the effects of lockdown, there is no description of COVID-19 nor of lockdowns. These descriptions must be added. The last weakness is also a limitation that remains unmentioned. That is, the role of the medical interns in interviewing the older adults. There is no description provided regarding the training the medical interns received in order to do the interviews and whether they used a standardized script. It remains unclear what the authors’ relationship was to these medical interns or in what way the authors supervised the data collection.

The authors should please note that, for MDPI journals such as IJERPH, the punctuation between two or more citations is a comma, not a semi-colon. There two instances (lines 323 and 327) that need correction in this regard. As well, abbreviated journals in the references should have a period after the abbreviation. Most of the journal references provided do not have these periods. Finally, do not include the journal number, just the journal volume, when referencing. There are a number of instances where the authors have included the journal number.

Line by line suggested edits.

22-23 “COVID-19 perceived susceptibility was not found to mediate the relationship between optimism and social support and depression or mediation use.” This statement is unclear. It seems to be saying there is a three-way relationship among optimism, social support and depression or a three-way relationship among optimism, social support and mediation use. If this is not what the authors intend, please provide clarity to the intended point.

35 Please add a line describing the COVID-19 virus. As well, add another line here indicating that lockdowns were considered necessary to allay COVID-19, as this information will be needed before mention of them in line 41

41 Change “base” to “basis”.

64-65 The authors are using Erikson’s psychosocial development theory and citing his 1982 book. They need to explain why they have selected Erikson’s model and demonstrate, by citing current references in peer reviewed journals, that what Erikson had to say about his final, 8th stage of psychosocial development is still considered relevant today. Please examine the following:

Gilleard C. The final stage of human development? Erikson's view of integrity and old age. Int. J. Ageing Later Life 202014, 139-62. https://doi.org/10.3384/ijal.1652-8670.1471

Van der Kaap-Deeder, J.; Soenens, B.; Van Petegem, S.; Neyrinck, B.; De Pauw, S.; Raemdonck, E.; Vansteenkiste, M. Live well and die with inner peace: The importance of retrospective need-based experiences, ego integrity and despair for late adults’ death attitudes. Arch. Gerontol. Geriatrics 202091, 104184. https://doi.org/10.1016/j.archger.2020.104184

69 Do the authors mean that the effects were not as adverse or that there were fewer adverse effects in number? If the latter, change “less” to “fewer”.

77 Change “Life events are as stressful as perceived as dangerous (primary appraisal) and perceived as lacking” to “The stressfulness of life events corresponds to the perception that they are dangerous (primary appraisal) and lacking”. 

86 Change “base” to “basis”.

113 Do the authors mean that the behaviors were less protective or that there were fewer protective behaviors? If the latter, change “less” to “fewer”.

120 Did the authors intend for COVID-19 to be italicized? Is this necessary? If not, please remove the italics.

144 Change “During this period” to “Over this period”.

146 What was the association between the authors and the family medicine interns? How were these family medicine interns recruited to participate in this study? How were they supervised by the authors? What training did they received to conduct the interviews. Please state the answers to these questions here in the manuscript.

147 What was the incentive for these older individuals to participate in this study? Please provide this information here in the manuscript.

156 If the scale was from 1 to 5, why in Table 1 are only three results for health status reported (Bad, Moderate and Good). How were these five options reduced to three? Either provide information regarding how and why the reduction was done or redo the table to list all five options of the replies provided.

224 Table 2: Please reduce the font size of “1.” to correspond to the rest of the numbering system of the Variables.

310 Change “Optimism and is” to “Optimism is”.

311 In lines 64-65, the authors state, “optimism may be paramount in older age”. Here, they state that it “is” paramount. Erikson does not say that it “is” paramount. Please adjust this sentence to say it may be paramount.

317 Change “Covid-19” to “COVID-19”.

342 The citation to a 2010 obstetrics and gynecology paper has little relevance here. Instead, this is a Google search of current publications regarding the Health Belief Model specific to older adults. The authors will note that almost all the papers resulting from this search are COVID-19 related. https://scholar.google.ca/scholar?hl=en&as_sdt=0%2C5&as_ylo=2019&q=Health+Belief+Model+older+adults&btnG=

352 This is the first time the reader learns that the response rate was high. This information should have been reported in line 132 when the total number of older adults who participated was mentioned. How many were contacted to participate and did not? Please state this here in the paper.

356 Another limitation of the study was that the participants were interviewed by more than one medical intern and not necessarily any of the authors. Therefore, without the participation/supervision of the authors, it cannot be known if the manner in which the medical interns interviewed the participants affected the type of answers the older individuals provided. This especially so since the authors don’t state that the medical interns used a standardized and well-rehearsed script when conducting the interviews.

364 Change “externa” to “external”.

366-367 “the findings support the cognitive model of coping with stressors [45]” This citation is to 1984 publication. The authors must demonstrate there is good reason to suppose that this model is still of relevance today by referencing current articles in peer reviewed journals.

Please examine the results of this Google search on this topic: https://scholar.google.ca/scholar?hl=en&as_sdt=0%2C5&as_ylo=2019&q=cognitive+model+of+coping+with+stressors+in+older+adults&btnG=

381 From the results of the data, low levels of education should also be included.

Author Response

Reviewer 2: This paper proposes two hypotheses related to older adults, COVID-19, and depression, (1) “higher optimism and higher perceived social support will be negatively associated with depression, while higher COVID-19 perceived susceptibility will be positively associated with depression, which in turn will be positively associated with more use of medication among older adults” and (2) “perceived susceptibility would mediate the relationship between psychosocial resources (optimism and perceived social support) and depressive symptoms and medication use.” To test these hypotheses, 383 older Israeli adults were interviewed by phone by medical interns to answer a series of questions constructed by the authors based on previous research in this regard. The results were that the first hypothesis was confirmed, but the second was not. The authors speculate that the reason for the unexpected result regarding the second hypothesis may be explained by the Health Belief Model. The study is seen to have both theoretical and practical implications related to psychosocial resources affecting older adults. It is considered the first study of its kind with respect to older adults.

The strengths of this manuscript are that it is clearly written, each of the measures used is properly referenced, the measures chosen were reasonable and directly relevant to the hypotheses to be investigated and the work is in an under studied area in need of research. The weaknesses are that the authors base important aspects of their research on old references without an investigation into whether the research conducted at that time still holds true today. This work will have to be done to improve the paper. Furthermore, although this is a paper directly related to COVID-19 and the effects of lockdown, there is no description of COVID-19 nor of lockdowns. These descriptions must be added. The last weakness is also a limitation that remains unmentioned. That is, the role of the medical interns in interviewing the older adults. There is no description provided regarding the training the medical interns received in order to do the interviews and whether they used a standardized script. It remains unclear what the authors’ relationship was to these medical interns or in what way the authors supervised the data collection.

Answer: We thank the Reviewer for the helpful suggestions for revising and improving our manuscript. We revised the paper with the aim of writing it in a more concise manner and made corrections as suggested by the Reviewer as follows:

Reviewer 2: The authors should please note that, for MDPI journals such as IJERPH, the punctuation between two or more citations is a comma, not a semi-colon. There two instances (lines 323 and 327) that need correction in this regard. As well, abbreviated journals in the references should have a period after the abbreviation. Most of the journal references provided do not have these periods. Finally, do not include the journal number, just the journal volume, when referencing. There are a number of instances where the authors have included the journal number.

Answer: These errors were corrected.

Line by line suggested edits.

Reviewer 2: 22-23 “COVID-19 perceived susceptibility was not found to mediate the relationship between optimism and social support and depression or mediation use.” This statement is unclear. It seems to be saying there is a three-way relationship among optimism, social support and depression or a three-way relationship among optimism, social support and mediation use. If this is not what the authors intend, please provide clarity to the intended point.

Answer: We have deleted this line as it is unnecessary and confusing.

Reviewer 2: 35 Please add a line describing the COVID-19 virus. As well, add another line here indicating that lockdowns were considered necessary to allay COVID-19, as this information will be needed before mention of them in line 41

Answer: In line with the Reviewer's comment, the following was added at the beginning of the Introduction:

Severe acute respiratory syndrome coronavirus-2 (SARS-CoV-2), the virus accountable for the COVID-19 pandemic, has extensively impacted the worldwide population [1]. Old age was affirmed as a risk for COVID-19 complications [2]; consequently, policymakers and media instructed older adults to confine social interactions [3].

References were amended in accordance:

Reviewer 2: 41 Change “base” to “basis”.

Answer: This was changed.

Reviewer 2: 64-65 The authors are using Erikson’s psychosocial development theory and citing his 1982 book. They need to explain why they have selected Erikson’s model and demonstrate, by citing current references in peer reviewed journals, that what Erikson had to say about his final, 8th stage of psychosocial development is still considered relevant today. Please examine the following:

Gilleard C. The final stage of human development? Erikson's view of integrity and old age. Int. J. Ageing Later Life 202014, 139-62. https://doi.org/10.3384/ijal.1652-8670.1471

Van der Kaap-Deeder, J.; Soenens, B.; Van Petegem, S.; Neyrinck, B.; De Pauw, S.; Raemdonck, E.; Vansteenkiste, M. Live well and die with inner peace: The importance of retrospective need-based experiences, ego integrity and despair for late adults’ death attitudes. Arch. Gerontol. Geriatrics 202091, 104184. https://doi.org/10.1016/j.archger.2020.104184

Answer: We thank the Reviewer for pointing out important point. As a result of this comment, together with the comments of Reviewer one, we have decided that in this particular paper the reference to Erikson's theory is unnecessary, therefore, we have decided to delete this.

Reviewer 2: 69 Do the authors mean that the effects were not as adverse or that there were fewer adverse effects in number? If the latter, change “less” to “fewer”.

Answer: This line was deleted due to the previous comment.

Reviewer 2: 77 Change “Life events are as stressful as perceived as dangerous (primary appraisal) and perceived as lacking” to “The stressfulness of life events corresponds to the perception that they are dangerous (primary appraisal) and lacking”. 

Answer: In line with the comments of Reviewer 1, we have completely deleted this section.

Reviewer 2: 86 Change “base” to “basis”.

Answer: We have changed this.

Reviewer 2: 113 Do the authors mean that the behaviors were less protective or that there were fewer protective behaviors? If the latter, change “less” to “fewer”.

Answer: This was changed to fewer.

Reviewer 2: 120 Did the authors intend for COVID-19 to be italicized? Is this necessary? If not, please remove the italics.

Answer: The Italics were removed

Reviewer 2: 144 Change “During this period” to “Over this period”.

Answer: This was changed.

Reviewer 2: 146 What was the association between the authors and the family medicine interns? How were these family medicine interns recruited to participate in this study? How were they supervised by the authors? What training did they received to conduct the interviews. Please state the answers to these questions here in the manuscript.

Answer: We have added to the manuscript the following:

The third researcher (M-A) recruited nine family doctors willing to distribute the questionnaires. The interns received training on conducting an interview from the researchers (S.S. and A.L.) who supervised the conduct of the interviews.

Reviewer 2: 147 What was the incentive for these older individuals to participate in this study? Please provide this information here in the manuscript.

Answer: There was no incentive, participation was on a voluntary basis. We added to the manuscript that participation was on a voluntary basis.

Reviewer 2: 156 If the scale was from 1 to 5, why in Table 1 are only three results for health status reported (Bad, Moderate and Good). How were these five options reduced to three? Either provide information regarding how and why the reduction was done or redo the table to list all five options of the replies provided.

Answer: We have provided the information for all five options in the Table.

Reviewer 2: 224 Table 2: Please reduce the font size of “1.” to correspond to the rest of the numbering system of the Variables.

Answer: This was corrected.

Reviewer 2: 310 Change “Optimism and is” to “Optimism is”.

Answer: This was corrected

Reviewer 2: 311 In lines 64-65, the authors state, “optimism may be paramount in older age”. Here, they state that it “is” paramount. Erikson does not say that it “is” paramount. Please adjust this sentence to say it may be paramount.

Answer: As mentioned above, we deleted any reference to Erikson from the manuscript. In addition, we changed the word paramount to essential.

Reviewer 2: 317 Change “Covid-19” to “COVID-19”.

Answer: This was changed.

Reviewer 2: 342 The citation to a 2010 obstetrics and gynecology paper has little relevance here. Instead, this is a Google search of current publications regarding the Health Belief Model specific to older adults. The authors will note that almost all the papers resulting from this search are COVID-19 related. https://scholar.google.ca/scholar?hl=en&as_sdt=0%2C5&as_ylo=2019&q=Health+Belief+Model+older+adults&btnG=

Answer: Thank you for pointing out important point. We have made reference to older adults and COVID-19 by replacing the reference with the following:

Bechard, L. E.; Bergelt, M.; Neudorf, B.; DeSouza, T. C., Middleton, L. E.. Using the health belief model to understand age differences in perceptions and responses to the COVID-19 pandemic. Front. Psychol 2021 12, 609893.

Reviewer 2: 352 This is the first time the reader learns that the response rate was high. This information should have been reported in line 132 when the total number of older adults who participated was mentioned. How many were contacted to participate and did not? Please state this here in the paper.

Answer: Approximately 5% of potential participants chose not to participate in the study. This information was added to the manuscript under the heading: Research population and sample

Reviewer 2: 356 Another limitation of the study was that the participants were interviewed by more than one medical intern and not necessarily any of the authors. Therefore, without the participation/supervision of the authors, it cannot be known if the manner in which the medical interns interviewed the participants affected the type of answers the older individuals provided. This especially so since the authors don’t state that the medical interns used a standardized and well-rehearsed script when conducting the interviews.

Answer: This was added to the limitations.

Reviewer 2: 364 Change “externa” to “external”.

Answer: This was corrected.

Reviewer 2: 366-367 “the findings support the cognitive model of coping with stressors [45]” This citation is to 1984 publication. The authors must demonstrate there is good reason to suppose that this model is still of relevance today by referencing current articles in peer reviewed journals.

Please examine the results of this Google search on this topic: https://scholar.google.ca/scholar?hl=en&as_sdt=0%2C5&as_ylo=2019&q=cognitive+model+of+coping+with+stressors+in+older+adults&btnG=

Answer: In accordance with the comments of Reviewer 1, this statement and reference was deleted from the manuscript.

Reviewer 2: 381 From the results of the data, low levels of education should also be included.

Answer: We added this to the conclusion.

Finally, numbers of all references cited in the text and the reference list were corrected in accordance with the changes suggested by both Reviewers.

We once again thank the Reviewer for the valued comments that helped improve our paper. We hope that it is now ready for acceptance.

*******************************************************

Round 2

Reviewer 1 Report

The authors made minor adjustments to the introduction. I went through all the sentences and most are exactly the same as the first version. My argument is supported by maintaining the objective in the first paragraph. In the response letter, the authors point out that my recommendations were met, but they only did so on one occasion or another. Thus, the study has a poor rationale, is free of major innovations and its fluency is confusing.

Author Response

Reviewer 1: The authors made minor adjustments to the introduction. I went through all the sentences and most are exactly the same as the first version. My argument is supported by maintaining the objective in the first paragraph. In the response letter, the authors point out that my recommendations were met, but they only did so on one occasion or another. Thus, the study has a poor rationale, is free of major innovations and its fluency is confusing.

Answer: We thank the Reviewer for your feedback and suggestions on our manuscript. As we may not have been clear about the changes we made in the first Round of corrections, we have now made a specific list of the corrections we made to the Introduction in Round one of the Review processes. Thereafter, we made another list of additional changes that we now made to improve the rational, highlight the innovation, and improve the fluency. We have accepted the original changes from Round 1, and have only left the corrections from Round 2 of the revision in red or deleted and highlighted in yellow.

Changes made to the Introduction in Revision Round 1

As recommended by the Reviewer we deleted all the subheadings and shortened all the paragraphs as follows:

Depression

Previous research underscores the prevalence of depressive symptoms in the older population, with approximately 10-30% of community-dwelling older adults reporting symptoms [9],

Based on Erikson's psychosocial development theory [14], optimism….

In contrast, pessimism has been linked with adverse results including depression and anxiety [18]

During the pandemic, optimistic undergraduate students with reduced pessimism reported less adverse effects of COVID-19 stress on depression [19]. Among older adults,……

Social support

According to the stress-buffering model, social support acts as a buffer via reliable relations and appraisals [21]. The stress-buffering model has five basic assumptions [21]: (1) Life events are as stressful as perceived as dangerous (primary appraisal) and perceived as lacking a suitable response (secondary appraisal). (2) Coping strategies determine the psychological outcomes of life events. (3) Social support is a resource that buffers stress via appraisals and coping. (4) Social support contains enacted support and perceived support. (5) Social support buffers stress when pertinent to the stressor [21]. 

Moreover, links were reported in studies from the HIV/AIDS, H1N1 influenza, SARS, and Ebola pandemics between social support and fewer mental health problems [23; 24].

Nevertheless, limited studies examined the stress buffering role of social support during the COVID-19 pandemic. Of these, perceived social support buffered the link between COVID-19 worries and psychological health among American college students [25]. To the best of our knowledge, the stress buffering model has not been explored

., and thus will be incorporated in the present study, together with the findings of a previous study in Israel, showing a link between higher optimism and social support with lower COVID-19 perceived susceptibility [11].

Perceived susceptibility

This was demonstrated in China during the SARS epidemic in 2003, with those with low perceived susceptibility to infection employing fewer personal hygiene practices such as hand sanitation and wearing face masks [32].

. Media exposure to COVID-19 news was also found to play a vital role in the public's rising perceived susceptibility and anxiety levels [35]. Finally,

The current study

In addition, we added and changed the following:

  1. Severe acute respiratory syndrome coronavirus-2 (SARS-CoV-2), the virus accountable for the COVID-19 pandemic, has extensively impacted the worldwide population [1]. Old age was affirmed as a risk for COVID-19 complications [2]; consequently, policymakers and media instructed older adults to confine social interactions [3].
  2. Optimism, known as mechanism for appraising one's life [15], may be essential in older age.
  3. During the pandemic, optimism and social support were found to buffer depression among older adults [12].
  4. On this basis, the social distancing and self-isolation policies of the COVID-19 pandemic may have adverse mental consequences due to the absence of social support [20],…..
  5. This relationship will be explored in the current study in the context of depression and medication use during the COVID-19 pandemic among older adults.
  6. Although health awareness increased during the COVID-19 pandemic [27], a lower perceived susceptibility of contracting COVID-19 was linked with fewer protective behaviors [28].
  7. Studies during the pandemic linked perceived susceptibility to COVID-19 and mental well-being, demonstrated by an association between social closeness to people infected by COVID-19 with higher perceived susceptibility and higher anxiety levels [29], and a high COVID-19 perceived susceptibility was linked with depression among older adults [12].
  8. Based on the above, the current study aimed to broaden the knowledge on the mental outcomes of stressful life events in old age, specifically the role of psychosocial resources (optimism and social support respectively) and COVID-19 perceived susceptibility to predict depression and the use of medication among older adults during the COVID-19 pandemic.

We have accepted all the above changes and will now refer to the changes made to the Introduction in Revision Round 2 (marked in the paper in red or deleted and highlighted in yellow).

  1. Line 38-40 of the Introduction the following was edited as follows:

Older age was established as a threat for COVID-19 complications [2], with studies during the pandemic showing that older adults displayed depressive symptoms, anxiety [5], peritraumatic distress [6], post-traumatic stress disorder [7], and depression even after receiving the vaccination [9].

  1. The following was added to line 43-49 of the Introduction:

Nevertheless, factors exploring adverse mental reactions together with an increase in medicine use among the older population during pandemics is limited. The American Geriatrics Society recently stated that knowledge on the benefits and harmful outcomes of medication use in older adults is frequently limited [10]; therefore, factors exploring increased medication use in this population is vital, particularly during stressful life events such as the COVID-19 pandemic.

Added reference to reference list:

  1. Lines 54-60 of the Introduction were changed as follows:

Depressive symptoms are perilous to older adults' overall health [10] [4], with studies reporting links between the COVID-19 pandemic and depression in older adults [2]. Beck’s [11] cognitive Developmental Model of Depression proposes that depressive symptoms embody from negative interpretations of life events. This was as demonstrated during the pandemic;, with a relationship found between older adults with holding negative world assumptions reported and a 4.4 times higher probabilities probability for clinical depressive symptoms than those with more positive world assumptions [8].

The following reference was deleted from the reference list:

Agustini, B.; Lotfaliany, M.; Mohebbi, M.; Woods, R.L.; McNeil, J.J.; Nelson, M.R.; Shah, R.C.; Murray A.M.; Reid, C.M.; Tonkin, A., Ryan, J., Williams, L.J.; Forbes, M.P.; Berk, M. Trajectories of depressive symptoms in older adults and associated health outcomes. Nat Aging 2022 2, 295-302. 

  1. The following line was added to lines 64-65 of the Introduction:
    Nevertheless, these effects have been less explored in terms of medication use among older adults.

  1. Lines 66-69 of the Introduction were changed as follows:

Optimism and pessimism play a vital role when coping with negative experiences [13], such as the COVID-19 pandemic. It has been suggested that people as optimistic people inclined to be optimistic strive when confronted with life's challenges, while those with a more pessimistic tendency outlook tend to withdraw [14].

  1. The following was changed and added to lined 75-77 of the Introduction:

Nevertheless, there is need for further study of this association among older adults on physical and mental health [12]., specifically, medication use during stressful life vents such as the COVID-19 pandemic.

  1. In order to further shorten the Introduction, the following was deleted from line 84-89:

On this basis, the social distancing and self-isolation policies of the COVID-19 pandemic may have adverse mental consequences due to the absence of social support [20], as demonstrated by studies around the globe during the COVID-19 pandemic [21,22,23]. For example, a German study showed a link between perceived social support and lower levels of anxiety, depression, and sleeping disorders [21]. Likewise, a Chinese study revealed a link between psychological distress with lower social support during the pandemic [22]. Similarly, a Spanish study found a link between living alone with adverse mental outcomes and sleep problems among older adults with mild cognitive impairment

  1. Likewise, the following was shortened in line 93-97 of the Introduction:

Perceived susceptibility describes the subjective threat of contracting a disease on a range from complete denial of any prospect of contracting the disease, to recognizing the statistical risk, and finally to perceiving that contracting the disease will occur [24]., which is known to be associated with increased It was found that perceived susceptibility increases depression in old age later life, especially during uncertain situations [5].

  1. Also, in lines 104-110 of the Introduction the following was shortened and changed:

Studies during the pandemic linked perceived susceptibility to COVID-19 and mental well-being [29], demonstrated by an association between social closeness to people infected by COVID-19 with higher perceived susceptibility and higher anxiety levels [29], and a high COVID-19 perceived susceptibility was linked with such as depression among older adults [12]. Nevertheless, few studies have explored the association between COVID-19 perceived susceptibility with depression this association needs further exploration among older adults, while this the relationship between perceived susceptibility and medicine use among older adults link remains unexplored. with regards to medicine use among older adults.

  1. Finally, the lines 113-115 of the Introduction were shortened and changed as follows:

Based on the above, the current study aimed to broaden the knowledge on the mental outcomes of stressful life events in old age, specifically the role of psychosocial resources (optimism and social support respectively) and COVID-19 perceived susceptibility to predict depression and the factors associated with depression and use of medication among older adults during the COVID-19 pandemic.

We once again thank the Reviewer for the kind assistance in helping us improve our manuscript. We hope that the manuscript is now acceptable for publication.